# Effect of the Ti6Al4V Alloy Track Trajectories on Mechanical Properties in Direct Metal Deposition

**Ivan Erdakov**, **Lev Glebov**, **Kirill Pashkeev, Vitaly Bykov, Anastasia Bryk, Vyacheslav Lezin and Liudmila Radionova** *

Department of Metal Forming, South Ural State University, Lenin Prospect 76, 454080 Chelyabinsk, Russia; erdakovin@susu.ru (I.E.); 79193293392@yandex.ru (L.G.); pashkeevki@susu.ru (K.P.); vitality.bykov.97@gmail.com (V.B.); 89193497818n@gmail.com (A.B.); vyacheslavlezin@gmail.com (V.L.)

* Correspondence: radionovalv@susu.ru; Tel.: +7-351-901-93-32

**Abstract:** The TiAl6V4 alloy is widely used in selective laser melting and direct laser melting. In turn, works devoted to the issue of how the track stacking scheme affects the value of mechanical properties is not enough. The influence of the Ti6Al4V alloy track trajectories on the microstructure and mechanical properties during direct laser deposition is studied in this article for the first time. The results were obtained on the influence of «parallel» and «perpendicular» technique of laying tracks in direct laser synthesis. All studied samples have a microstructure typical of the hardened two-phase condition titanium. Here, it is shown that the method of laying tracks and the direction of load application during compression testing relative to the location of the tracks leads to a change in the ultimate strength of the Ti-6Al-4V alloy from 1794 to 1910 MPa. The plasticity of the Ti-6Al-4V alloy obtained by direct laser alloying can vary from 21.3 to 33.0% depending on the direction of laying the tracks and the direction of the compression test. The hardness of alloys varies in the range from 409 to 511 HV and depends on the method of laying the tracks and the direction of hardness measurements.

**Keywords:** additive technology; titanium alloy; TiAl6V4; microstructure; tensile strength; direct metal deposition

## 1. Introduction

Titanium and its alloys are widely used in high-tech industries: Aircraft and rocketry, shipbuilding, nuclear power, chemical industry, and medicine.

Titanium and titanium alloys are better than the majority of modern construction materials, such as steel and aluminum, in their physical and mechanical properties and manufacturability. Titanium and titanium-based alloys are characterized by high melting points and electrical resistivity, their strength is comparable to most grades of alloy steels, they have high corrosion resistance in air and water, and a chemically aggressive environment. They are non-magnetic and have many other useful properties. The titanium specific mass is 56% of the specific mass of steel, which is easily processed by pressure and has biological inertness [1].

Adding other metals to titanium makes it possible to obtain alloys with a certain level of mechanical and operational properties. Their classification is divided into three groups:

(1) α-Structure alloys. This group of alloys does not have an increase in brittleness with prolonged exposure to high stresses and temperatures.

(2) α + β-Structure alloys. The β-phase is more ductile, the alloys are more workable and better amenable to pressure processing than α-structure alloys.

(3)　　β-Structure alloys. This group of alloys has good weldability, good technological ductility, and high strength.

　　The alloying elements are divided into three groups:
α-Stabilizers: Aluminum, oxygen, nitrogen, carbon;
β-Stabilizers: Vanadium, molybdenum, iron, chromium, nickel;
Neutral hardeners: Zirconium, tin, silicon [1].

　　The problem of melting titanium arises together with the problem of casting and, above all, casting molds. Casting of titanium in small quantities of samples or parts that are simple in shape can be carried out in cold copper, aluminum, or bronze molds. Heating the mold causes an interaction between the melt and the mold material. Mold filling difficulties arise with very rapid cooling, which to some extent, can be overcome by centrifuging and press molding. The technology of press molding was used for the manufacture of products from titanium and alloys previously. It was followed by complex and expensive processing, since titanium actively interacts with oxygen at high temperatures, which is necessary to provide a protective atmosphere of argon [2,3].

　　Titanium and its alloys are amenable to pressure treatment by all known methods: Forging, rolling in a cold and hot condition, stamping, and drawing. Its processing is similar to the stainless steel processing, but the modes depend on the amount and nature of impurities [1].

　　Titanium is amenable to cutting similar to stainless steel. High-speed steels and carbide are used as tool materials for titanium cutting. It is especially important in processing titanium by cutting to monitor the sharpness of the tool, since otherwise the processed material surface is crushed and its hardness increases greatly, which complicates further processing. Milling is also possible at medium and low speeds.

　　The main nuance of titanium and its alloys processing is the desired continuous processing. However, local hardening occurs in the material with a local increase in hardness since an increase in friction, due to a blunt or broken tool, leads to great difficulties in further processing. Titanium also tends to gall the tool when the turnings are overheated, which leads to tool breakage [1].

　　Titanium welding requires careful protection against contamination by oxygen and nitrogen not only for the molten metal in the welding zone, but for highly heated parts, including the opposite side of the welding seam. Heat treatment is recommended after welding [3].

　　It is necessary to use a large billet, on which long work is carried out with a large amount of waste for the manufacture of complex shape parts. Waste amounts to about 70–75% of the billet volume in some cases. In addition, parts of the complex construction that cannot be manufactured in one piece have to break down into simpler parts for further assembly.

　　Additive technologies are used to create complex products with non-standard geometry [3–5] at present, thereby making it possible to manufacture products billet as close as possible to the finished products with only needing to finish machining [6–8]. The topological optimization of parts [9], manufactured by additive methods, allows us to preserve all the strength characteristics of a traditional part, but less expensive material and less time will be invested in its production. Therefore, for example, in aircraft construction, it becomes possible to increase the flight distance of an aircraft using the same amount of fuel as before the topological optimization, or to reduce fuel costs with a decrease in the impact on the ecological environment due to a decrease in the weight of products. It is possible to significantly reduce the amount of waste by creating already, in close to real geometric form, shapes and sizes of the billet with 10–15% of waste using additive technologies [10].

　　Products from titanium alloys are manufactured using both selective laser melting (SLM) [11–24] and laser metal deposition (LMD) [25–27] technologies.

　　Selective laser melting is a process of layer-by-layer powder material decomposition in a tight chamber and laser sintering of a material layer along the product contours. Non-critical products made with SLM can meet the technical requirements of surface roughness even without finishing (turning and grinding). This can be achieved by selecting the process parameters such as the powder material layer thickness and the parameters of the laser radiation [12,13]. This process is applicable more for

small-scale products of small and medium size, with the possibility of creating closed cooling channels, which cannot be done by traditional methods of parts processing [15–19]. It is also possible to create net structures for facilitation, but with effective strength characteristics preservation of parts [21–24]. In addition, it is actively used to create prostheses, producing a specially porous structure for better melting of the prosthesis with bone tissues [24].

Laser metal deposition (LMD) is a process of direct material deposition under a constant source of laser treatment with the laser device movement along the product contours. It is possible to use several types of materials: Metal powder or wire in this method. The LMD process is applicable to create infinitely large products of complex geometric shapes with small tolerances and high properties [25]. Moreover, the process of direct laser deposition is used to create new components of parts with high geometric complexity [26], which would require much more production hours or the possibility of creating a single whole product using detachable components to create it using traditional processing methods. Earlier, we carried out work [26] on growing a special item of parabolic shape by combining the selective laser melting and direct metal deposition. There is no critical thermal effect on the entire remaining part due to spot heating by this method, in which the cost of creating a new product is reduced [27]. The direct laser melting process allows the gradient metal composites creation by mixing powders of different compositions in growing a product continuous process, thereby making it possible to obtain products that have no analogues on the market. Aircraft and machine builders use direct laser coating to repair high-value products.

Many disadvantage characteristics of traditional titanium and its alloys processing methods during the layer-by-layer alloying of powder material are absent, but they are replaced by other characteristic properties of additive technologies. The motion vector of the laser beam in the XYZ coordinate system during the layer-by-layer powder alloying in different components of the part changes repeatedly during the complex shape synthesis [20,22].

Defects, mainly microcracks and pores, may be present in products obtained by the selective laser melting technology [13]. Pores have a significant effect on the mechanical properties of the material due to their larger size and flat shape [15,16,19].

The porosity does not exceed 1% with direct laser melting, but the movement of the laser along different trajectories, with the possibility of texture formation in different metallographic planes [25–27], leads to the appearance of properties anisotropy [25].

The analysis of publications shows that the question of understanding the microstructure formation patterns and, as a consequence, the mechanical properties formation, depending on the scheme of layers deposition from a powder material during direct laser synthesis, remains unexplored. The relevance of such studies is determined by the need in developing effective additive processes for the production of complex shape products from the titanium alloy Ti6Al4V, which is widely demanded in the aviation industry.

The object of this work is to study the effect of the powder material deposition trajectory Ti6Al4V alloy on the microstructure and mechanical properties of the samples during direct laser synthesis.

## 2. Materials and Methods

The studies on the samples grown using the direct metal deposition (DMD) technology were carried out in the laboratories of the South Ural State University, Chelyabinsk, Russia.

The studies were carried out on a FL-Clad-R-4 laser complex (Figure 1) equipped with a 4 kW laser LS-4, a 6-axis industrial robot, a KUKA R-120 manipulator with a KUKA DKP-400 2-axis positioner, a TWIN-10-CR-2 powder feeder, as well as technological equipment for creating an argon atmosphere.

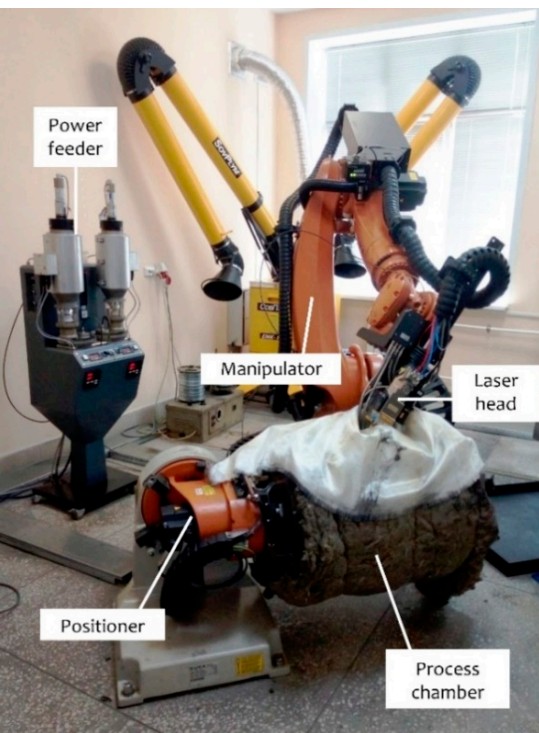

**Figure 1.** Laser complex FL-Clad-R-4.

Titanium powder Ti-6Al-4V with a fraction of 30–100 μm was chosen to carry out the research (Figure 2). The melting of the powder was conducted according to the previously obtained mode [24]: Power $P$ = 1800 W; powder feed K = 27 g/min; laser movement speed U = 25 mm/s; displacement of the track in the layer plane Δl = 2.5 mm; track displacement in the vertical plane Δh = 0.3 mm; number of layers $n$ = 150. Two types of powder deposition techniques have been developed. The layers were coated on each other by parallel tracks in the first technique (Figures 3a and 4a), and the layers parallel tracks were alternately shifted by 90 degrees in the second type (Figures 3b and 4b). The diameter of the titanium disk on which the sample was deposited was 54 mm. The overall dimensions of the obtained samples did not exceed 50 × 25 × 40 mm.

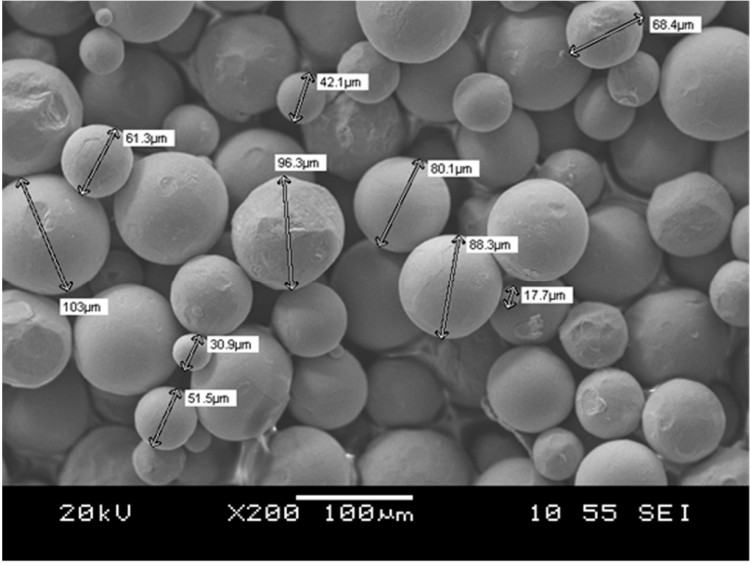

**Figure 2.** Determination of powder particle size with a scanning electron microscope.

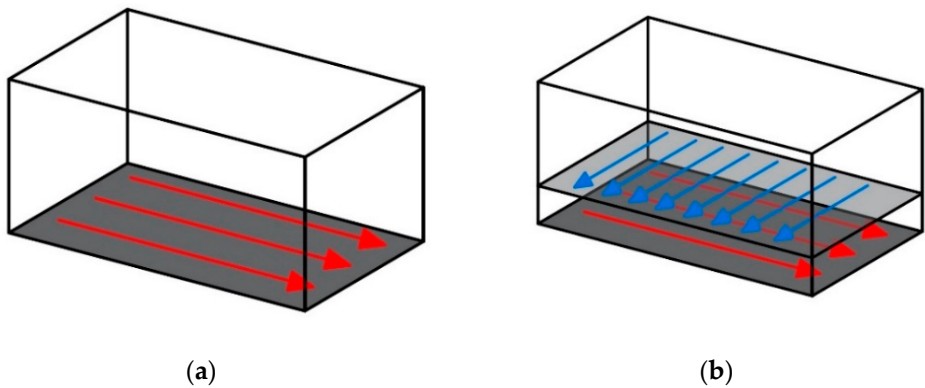

**Figure 3.** Motion tracks of the laser head during powder melting: (**a**) Sample no. 1, the "parallel" technique; (**b**) sample no. 2, the "perpendicular" technique.

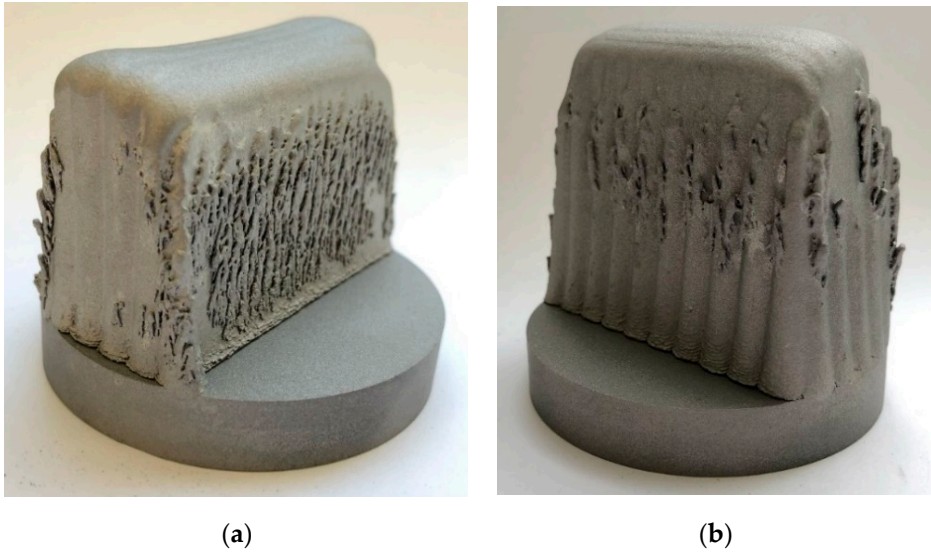

(**a**)                                  (**b**)

**Figure 4.** Samples obtained by the direct metal deposition (DMD) technology: (**a**) Sample no. 1, the "parallel" technique; (**b**) sample no. 2, the "perpendicular" technique.

The chemical composition of the samples structural components was analyzed on a JSM-6460LV scanning electron microscope (JEOL, Tokyo, Japan) equipped with an energy dispersive spectrometer (OxfordInstruments, Abingdon, UK) for qualitative and quantitative X-ray microanalysis. The microstructure of the samples was studied on the Axio Observer D1.m optical inverted metallographic microscope (Carl Zeiss Microscopy GmbH, Jena, Germany) equipped with the ThixometPro software (Thixomet Pro, Thixomet Company, Saint Petersburg, Russia).

Samples for mechanical tests were prepared using electrical discharge cutting (Figure 5a). This was necessary since the hardness of the obtained samples is high, which complicates their mechanical processing. Electric discharge machining allows excluding the heating of the metal in the cut zone and obtaining a high quality of the samples surface, which is very important for brittle materials mechanical testing.

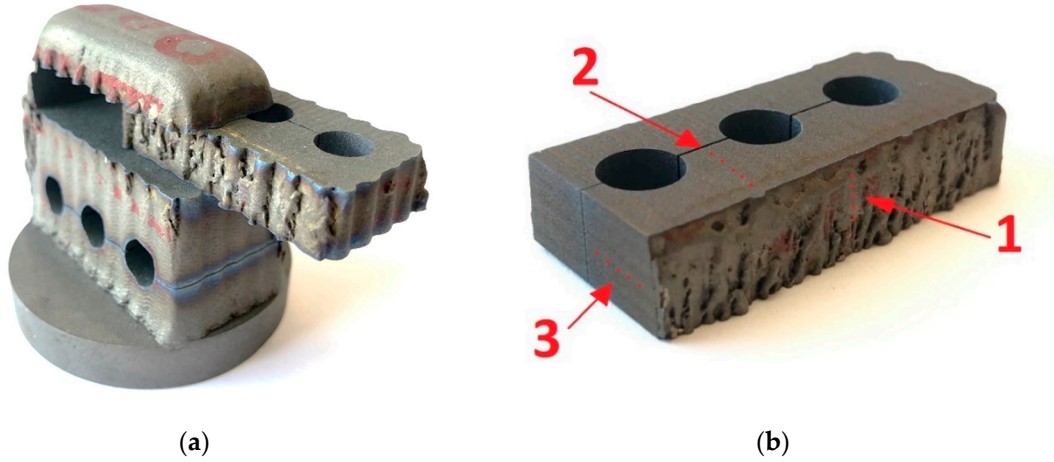

(**a**)　　　　　　　　　　　　　　　　　　　　　　　　(**b**)

**Figure 5.** Cut-up sketch for the electric discharge sawing to determine mechanical properties (**a**), obtaining samples for compression testing in mutually perpendicular directions of track formation; (**b**) obtaining samples for measuring hardness in planes 1, 2, and 3.

The hardness was determined using the Vickers scale on a hardness tester HV-1000 according to GOST 2999-75. The determination of hardness and the study of the microstructure was carried out along three planes, according to the directions indicated by the arrows in Figure 5b. Hardness tests were carried out on the same specimens on which the microstructure was studied. The hardness was measured at 5 points (Figure 5b) for each sample and the average was determined.

The compression test pieces were 10 mm in diameter and 20 mm in height. Compression tests were carried out on a universal testing machine INSTRON 5882, compression speed was 2 mm/min according to GOST 25.503-97. Compression tests were repeated three times for each type of samples.

Since specimens for compression testing were made according to the scheme in Figure 5a and for specimen no. 1, the "parallel" technique, and for specimen no. 2, the "perpendicular" technique, then four types of specimens for testing are obtained with the layout of the tracks, as shown in Figure 6.

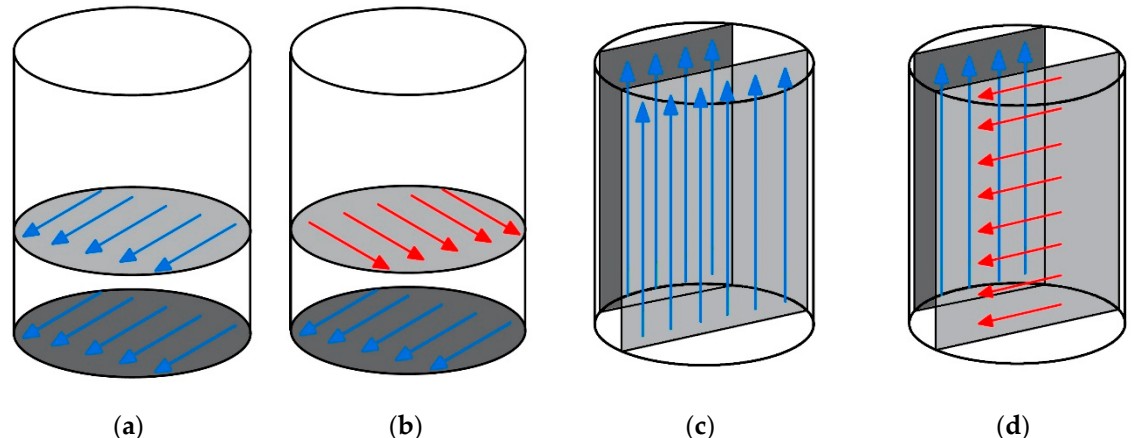

(**a**)　　　　　　　　(**b**)　　　　　　　　(**c**)　　　　　　　　(**d**)

**Figure 6.** Layout of the tracks in samples for compression tests: (**a**) "Parallel"; (**b**) "perpendicular"; (**c**) "vertical"; (**d**) "mixed".

## 3. Results

Both samples externally show characteristic tracks inherent in different growing methods. In turn, there are no visually detected cracks and open pores on the grown samples (Figure 4). Particles of over-fed powder form on the surface of the samples. The powder forms droplets or dendrites due to the high temperature.

The study of the microstructure of the obtained samples showed that a mechanical mixture of α- and β-phases is formed as a result of melting (Figures 7 and 8). Earlier, we carried out work [26] in which we studied, in more detail, the phase composition of the samples obtained from the powder Ti-6Al-4V using similar fusion modes. The presence of two phases (α and β) in the structure is confirmed by XRD results.

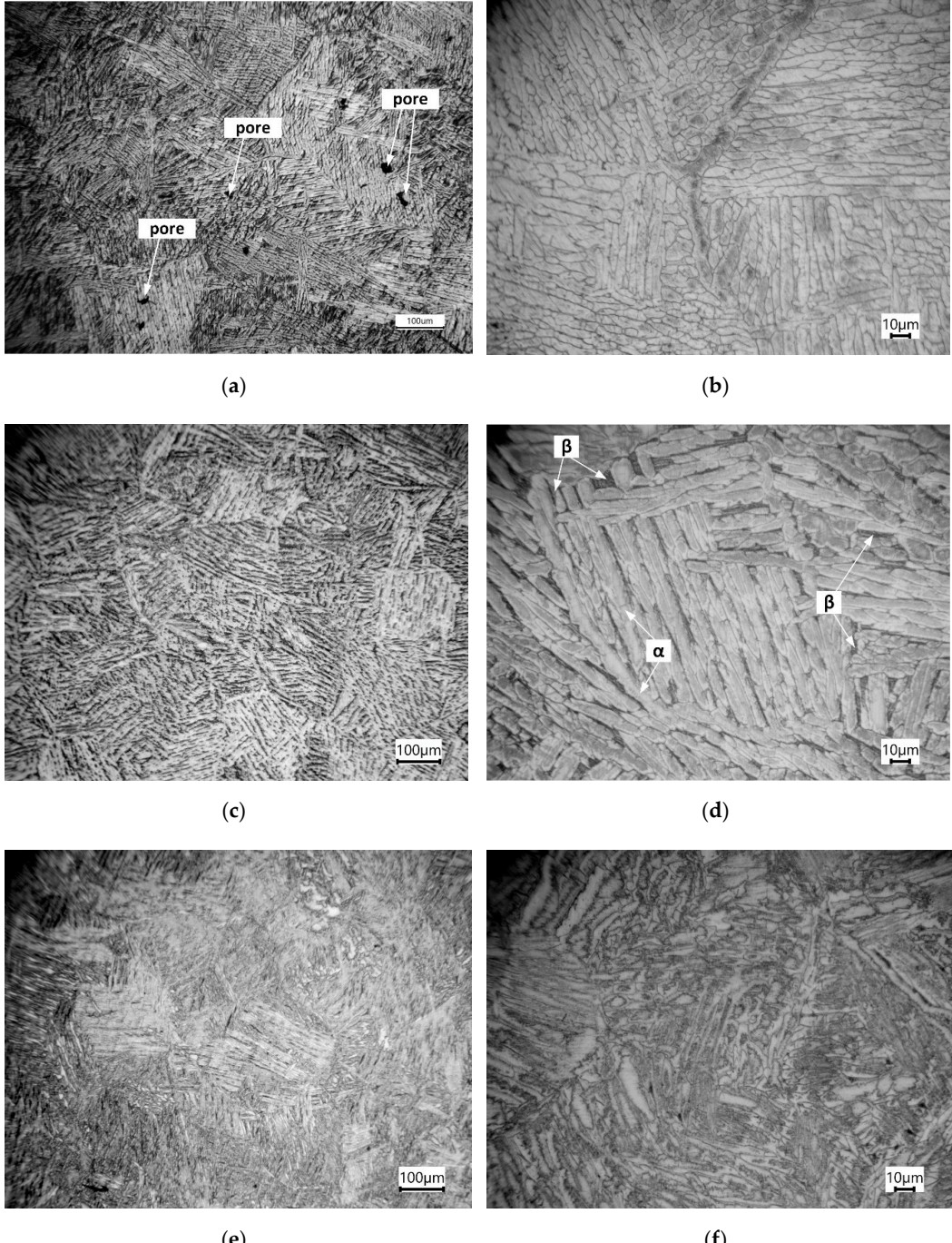

**Figure 7.** Microstructure of sample no. 1, the «parallel» technique: (**a**,**b**) Direction 1 according to Figure 5b; (**c**,**d**) direction 2; (**e**,**f**) direction 3.

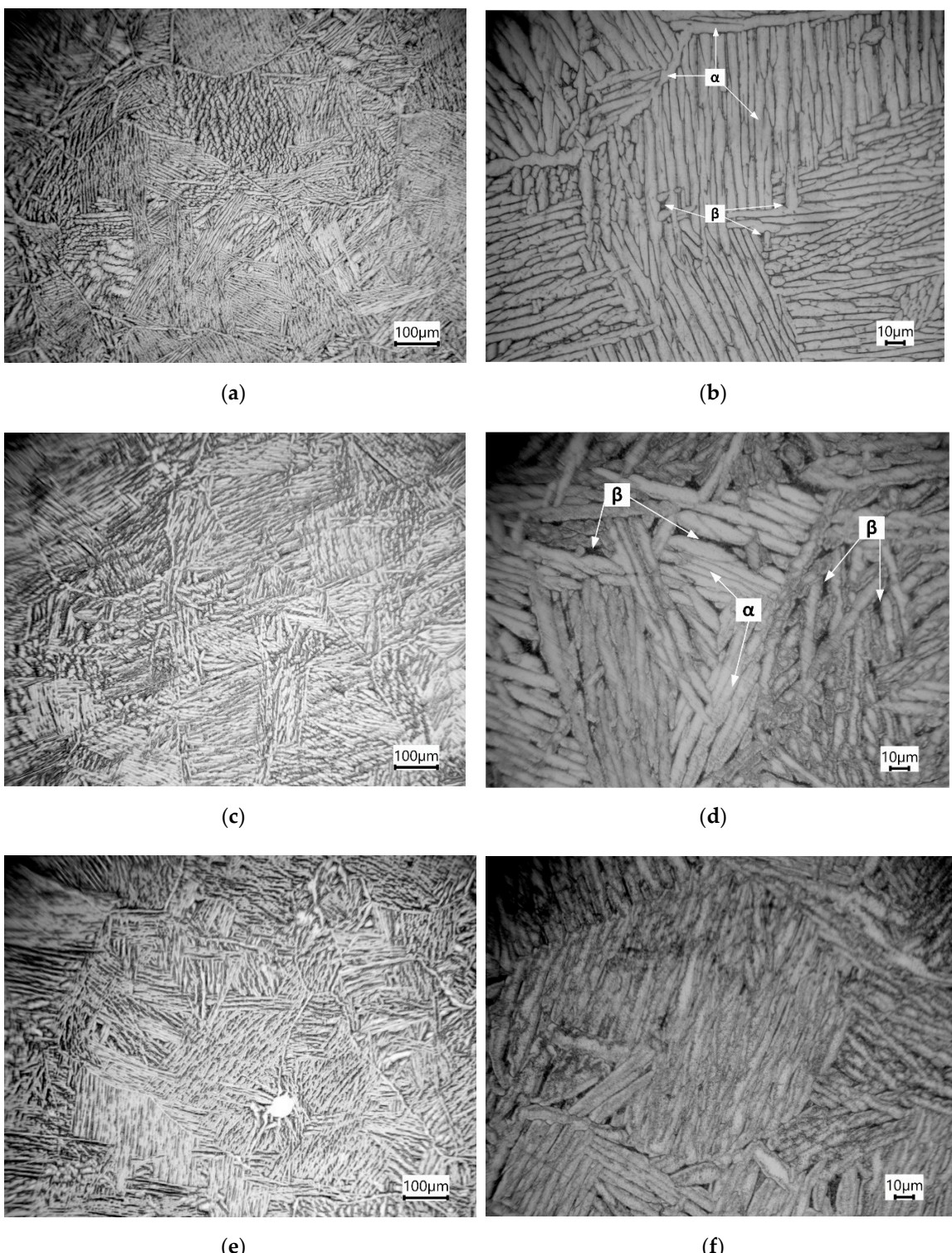

**Figure 8.** Microstructure of sample no. 2, the «perpendicular» technique: (**a**,**b**) Direction 1 according to Figure 5b; (**c**,**d**) direction 2; (**e**,**f**) direction 3.

The two-phase structure forms upon heating and fast cooling. Aluminum in an amount of 6% allows the formation of a long-range order $\alpha$-solid solution phase with the possibility of pre-separation of the $\alpha_2$-phase in the Ti-6Al-4V alloy. Vanadium is an isomorphic $\beta$-stabilizer, does not form chemical compounds with titanium, and does not lead to eutectoid reactions. The grains are formed independently of the applied layers, passing from one layer to another. The coarse-grained structure is

formed during the powder melting process. It should be noted that the number of pores is very small, and their size does not exceed 20 μm (Figures 7a and 9a). All studied samples have a microstructure typical of the hardened state of two-phase titanium. The so-called "basket weave" structure is clearly visible in some fields of view: The α-phase stands out along the grain boundaries of the primary β-phase, the acicular α-phase in the grain body with thin layers of the β-phase between them (Figure 8b,d).

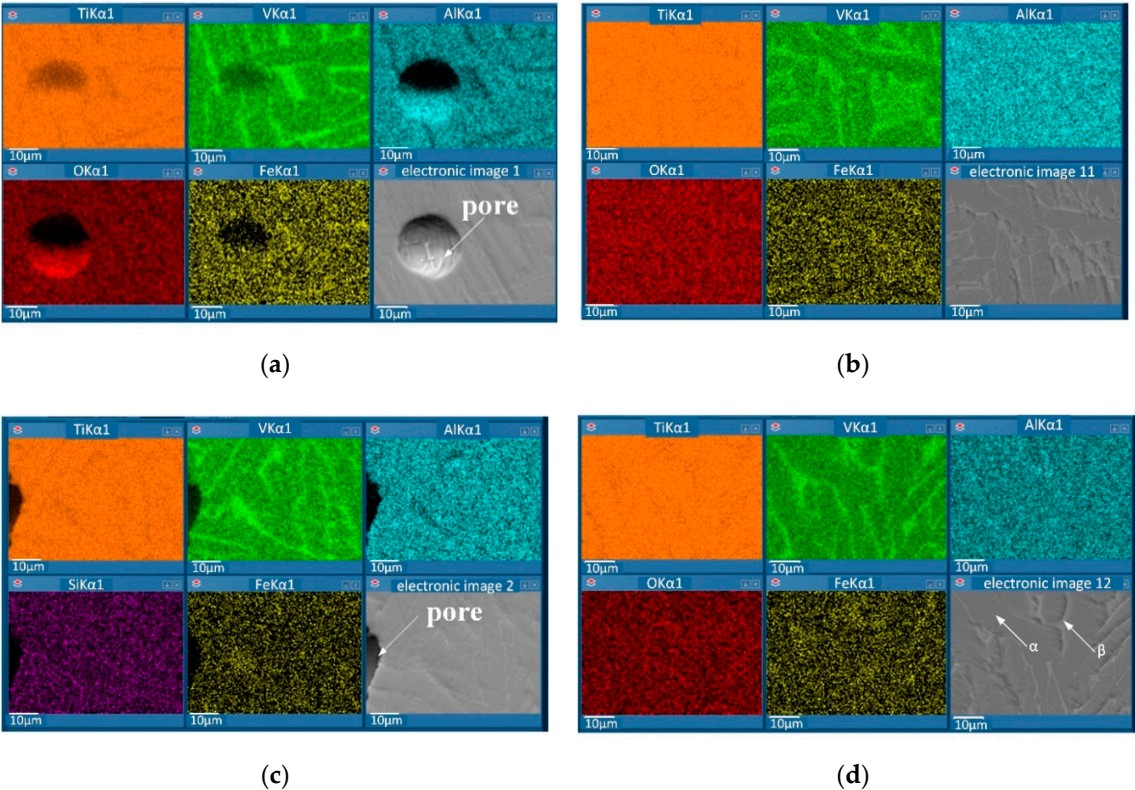

**Figure 9.** Energy-dispersive X-ray spectroscopy (EDS) mapping of the Ti-6Al-4V alloy samples: (**a**) Sample no. 1, the "parallel" technique; (**b**) sample no. 1, the "parallel" technique without pores; (**c**) sample no. 2, the "perpendicular" technique at times; (**d**) sample no. 2, the "perpendicular" technique without pores.

The results of studying the samples chemical composition using a scanning electron microscope are shown in Table 1 and Figure 9. Figure 9 shows the maps of the distribution of chemical elements in the samples under study. Areas of solid material without pores (Figure 9b,d) and areas with pores (Figure 9a,c) were studied. It can be seen from the results that the areas with pores do not differ in composition from the homogeneous area of the samples under study. The porosity is not localized. There are separate small spherical pores, the total percentage of porosity does not exceed 0.2%. The pore sizes do not exceed 20 μm. Pores are undesirable in this study. The reason for their formation is the shielding gas argon, which did not have time to escape when the powder was melted. The distribution of chemical elements in the field of observation is uniform. The only exception is V, the concentration of which is increased in the β-phase areas.

**Table 1.** Results of samples chemical analysis.

| Mass% | Al | Si | Ti | V | Fe | Amount |
|---|---|---|---|---|---|---|
| Sample no. 1, the «parallel» | 5.91 | 0.13 | 89.09 | 4.66 | 0.22 | 100.00 |
| technique | 5.98 | 0.09 | 89.15 | 4.58 | 0.20 | 100.00 |
| Sample no. 2, the | 5.86 | 0.11 | 89.25 | 4.64 | 0.15 | 100.00 |
| «perpendicular» technique | 5.92 | 0.12 | 89.48 | 4.29 | 0.18 | 100.00 |

The results of the hardness measurements in three directions (Figure 5b) of the samples grown by two methods are shown in Table 2. The hardness of the alloy depends on the tracks applying technique during the samples growth. Therefore, the «perpendicular» technique of tracks applying during melting allows increasing the hardness in comparison with the hardness of the samples obtained by the «parallel» technique. The largest increase in hardness is observed in one measurement direction and it is 36 HV or 8%. In turn, it should be noted that in both samples the value of hardness significantly depends on the direction of measurement. Therefore, the minimum value of hardness in samples 1 and 2 corresponds to direction 1, and the maximum to direction 3.

**Table 2.** Results of hardness measurements.

| Sample Type | Measurement Direction | Hardness Value, HV | | | | | Mean Value |
|---|---|---|---|---|---|---|---|
| | | 1 | 2 | 3 | 4 | 5 | |
| Sample no. 1, the «parallel» technique | 1 | 386 | 442 | 401 | 401 | 413 | 409 |
| | 2 | 449 | 475 | 486 | 494 | 460 | 472 |
| | 3 | 490 | 533 | 510 | 502 | 486 | 504 |
| Sample no. 2, the «perpendicular» technique | 1 | 435 | 467 | 428 | 449 | 446 | 445 |
| | 2 | 498 | 478 | 467 | 478 | 460 | 476 |
| | 3 | 482 | 536 | 510 | 519 | 510 | 511 |

The results of compression tests are presented in Table 3 and Figure 10. The maximum strength of 1910 MPa was shown by samples with a «vertical» track arrangement, and the minimum 1794 MPa with a «parallel» one. The alloy can resist higher stresses when a load is applied perpendicular to the track arrangement than when it is loaded parallel to the tracks. In turn, the deformation under compression for a sample with a «vertical» track arrangement is higher than for a sample with a «parallel» one. The "vertical" tracks arrangement provides a combination of high strength with high plasticity of the material. The highest plasticity, namely 33%, was shown by samples with a "mixed" track arrangement at the same time.

**Table 3.** Average values of mechanical properties depending on the layout of the tracks.

| Layout of Tracks in the Sample | Compressive Strength Rs, MPa | Young's Modulus, MPa | Compression Strain, % |
|---|---|---|---|
| «parallel» | 1794 | 17642 | 21,3 |
| «perpendicular» | 1847 | 17678 | 21,8 |
| «vertical» | 1910 | 16998 | 26,8 |
| «mixed» | 1817 | 15697 | 33,0 |

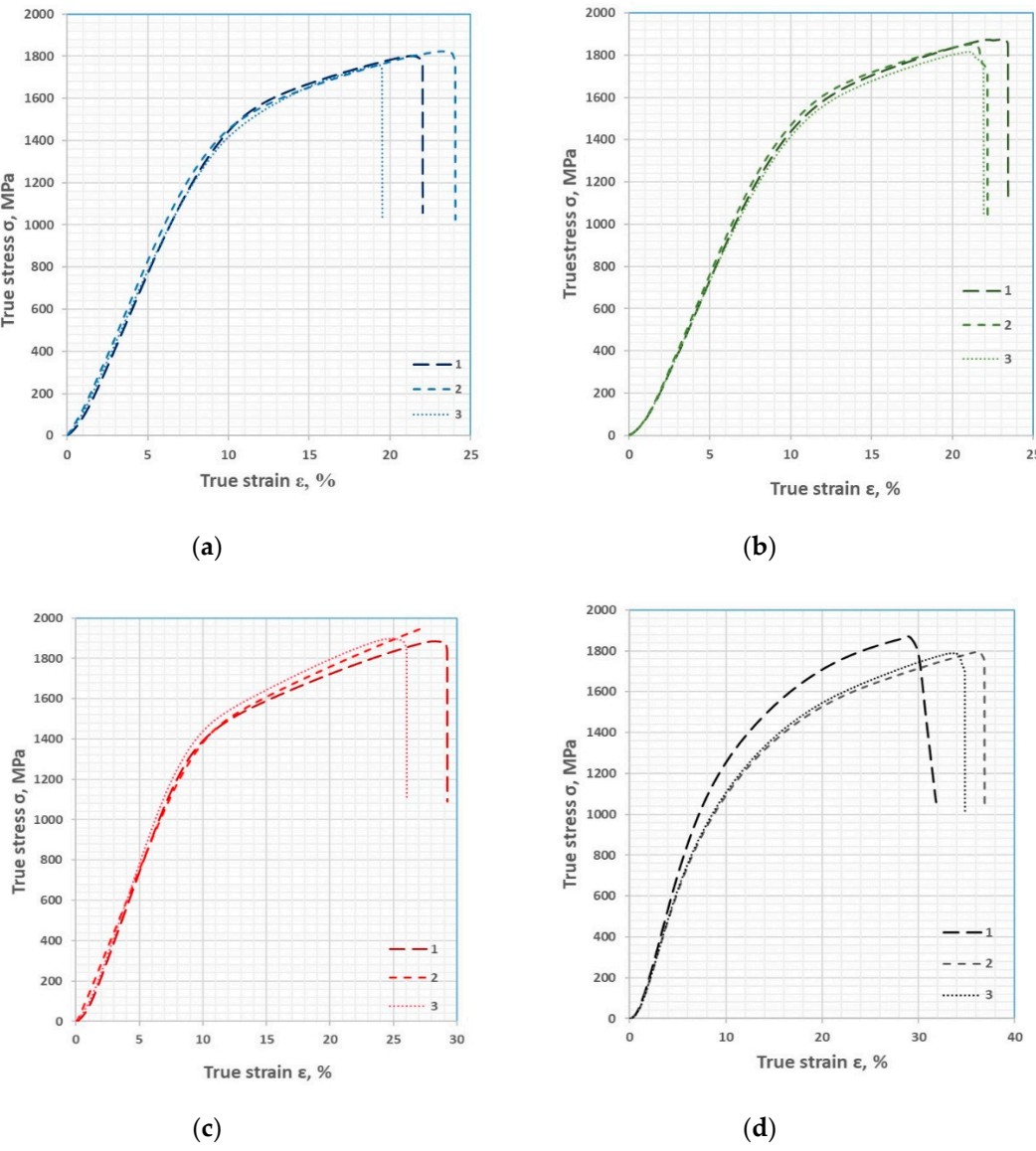

**Figure 10.** Compression diagrams of samples with different track layouts: (**a**) "Parallel"; (**b**) 6—"perpendicular"; (**c**) "vertical"; (**d**) "mixed"; 1,2,3—sample test number.

## 4. Discussion

It was shown that the laser complex FL-Clad-R-4 makes it possible to obtain billet or finished products [26] from titanium alloys by fusing the powder material in a protective argon atmosphere in the course of research [2,3]. A homogeneous microstructure from the Ti-6Al-4V alloy, which consists of a mechanical mixture of α- and β-phases (Figures 7 and 8), is formed when fusing a powder material with a fraction of 30–100 μm. The porosity of the obtained samples does not exceed 0.2%, and the size of spherical pores is less than 20 μm (Figures 7 and 9). The chemical analysis of the obtained samples showed that the alloy contains 89% Ti, 5.9% Al, and 4.3 to 4.7 V (Table 1). Vanadium is distributed unevenly, and its increased concentration leads to the formation of the β-phase in the microstructure (Figures 7–9). In turn, the presence of pores does not lead to a change in concentration, and the reason for their appearance is most likely argon gas bubbles. The melting of samples using «parallel» and «perpendicular» track arrangements affects the mechanical properties of alloys (Figure 10). Tests of samples made in different directions relative to the melting tracks showed the influence of the direction on the mechanical properties. The maximum strength of 1910 MPa was

shown by samples with a «vertical» track arrangement, and the minimum 1794 MPa with a «parallel» one. The highest plasticity, namely, a compression deformation of 33%, was shown by samples with a «mixed» track arrangement (Table 3). The hardness of the obtained samples varies in the range from 409 to 511 HV (Table 2) and also depends on the direction of measurement and the method of laying the tracks as «parallel» or «perpendicular». The reason for the pronounced anisotropy of properties is directional crystallization and subsequent recrystallization when the next layer is applied. Subsequent recrystallization is confirmed by the formation of grains larger than the track height. However, a more detailed explanation of the recrystallization mechanism and the dependence of properties on the direction of crystal growth requires targeted research that is planned for the future.

## 5. Conclusions

The influence of the Ti6Al4V alloy powder material deposition track on the microstructure and mechanical properties, during direct laser synthesis, is studied in this article for the first time. The study of the microstructure of the obtained samples showed that as a result of melting, a mechanical mixture of α- and β-phases is formed. All studied samples have a microstructure typical of the hardened two-phase condition titanium. The so-called "basket weaving" structure is clearly visible in some fields of view: The α-phase stands out along the grain boundaries of the primary β-phase, the needlelike α-phase in the grain body with thin layers of β-phase between them. The studies carried out have shown that it is necessary to take into account the layout of the tracks during direct laser melting of the powder material when designing finished parts of complex shapes or billet requiring finishing machining. It is shown that the «parallel» and «perpendicular» track arrangement is reflected in the mechanical properties of the material using the Ti-6Al-4V alloy as an example. The direction of the load application during the mechanical properties testing also has a great effect. It is possible to increase the value of the ultimate strength that the Ti-6Al-4V alloy resists from 1794 to 1910 MPa only by taking into account the location of the tracks relative to the applied compression load. In turn, the ductility of the Ti-6Al-4V alloy obtained by direct laser melting can change from 21.3 to 33.0% depending on the direction of laying the tracks and the direction of the compression test. The hardness of alloys varies in the range from 409 to 511 HB and also depends on the method of laying the tracks and the direction of hardness measurements. Taking these factors into account when designing critical parts from titanium alloys will make it possible to correctly perform calculations and prevent mistakes. It is expedient to investigate the effect of the Ti6Al4V alloy powder material deposition track when testing samples for static tension in the future.

**Author Contributions:** I.E. conducted the literature review and drew up the research plan, L.G. processed the experimental data and analyzed the results obtained, K.P. conducted the experimental research, V.B. conducted the experimental research, A.B. conducted the metallographic analysis, V.L. prepared the manuscript and L.R. reviewed the literature research, processed experimental data and formulated conclusions. All authors have read and agreed to the published version of the manuscript.

**Funding:** The APC was funded by Medelt, LLC 25a-20, Moldavskaya Street Chelyabinsk 454021 Russia VAT: 7451300610.

**Acknowledgments:** The work was carried out with the financial support of the Ministry of Science and Higher Education of the Russian Federation, within the framework of a subsidy for financial support for the fulfillment of a state task (fundamental scientific research), contract no. FENU-2020-0020 (2020071GZ).

**Conflicts of Interest:** The authors declare no conflict of interest.

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
