# Peer review of "Effect of the Ti6Al4V Alloy Track Trajectories on Mechanical Properties in Direct Metal Deposition"

_machines, doi:10.3390/machines8040079_

Round 1

Reviewer 1 Report

The paper can be published without any corrections.

Author Response

Dear reviewer, thank you for appreciating our work.
In the new version of the article, the "methods" section has been improved.

Reviewer 2 Report

The authors have used six-axis industrial robot to print titanium alloy with powder-based material in an inert chamber. The concept is quite innovative for making billet. However, there are some concerns and details comments are below.

1) In the introduction part author has talked in detail about the traditional manufacturing process and titanium material. The author must focus the introduction in additive manufacturing technology mainly on DMD and SLM. Challenges associated with previous technology and author novel solution for this. 

2) In material and methods section author talks about two processes of deposition parallel and perpendicular. However, in Figure 8 there are 4 methods. What was the layer thickness of the printed sample? The author must explain the details about the compression testing, what kind of compression was carried out and the direction of compression. Was there any ASTM standard followed? What is the size of the sample? 

3) In the result section alpha and beta phases should be marked in the picture. In figure 8a, There is a pore in parallel techniques and in 8b there are no pores. Was these pores were made for any purpose? or it is a defect?  If this is a defect it should also be marked in the SEM graph (Fig 6 and Fig 7). 

4) In the result section table 3 shows that the compressive strength of vertical tracks is higher and followed by perpendicular. The author must explain the reason behind this is discussion section along with the direction of the applied load. How the hardness value was checked must be explained using the picture. why should hardness value change due to different direction ? 

5) Figure 9 graph should be more clear. 

6) Overall the study must be replanned and presented in a better way. 

Author Response

Dear reviewer, thank you for appreciating our work.

We have done some work to improve the manuscript taking into account your comments.

1) In the introduction part author has talked in detail about the traditional manufacturing process and titanium material. The author must focus the introduction in additive manufacturing technology mainly on DMD and SLM. Challenges associated with previous technology and author novel solution for this.

In the introductory part, more than 50% of the text is devoted specifically to SLM and DMD technologies. Currently, there are no publications devoted to the influence of the direct growth trajectory of products and samples from powder materials. And this is very important to know when growing parts for designers. Therefore, this study is the beginning of a series of works that we began to carry out.

2) In material and methods section author talks about two processes of deposition parallel and perpendicular. However, in Figure 8 there are 4 methods. What was the layer thickness of the printed sample? The author must explain the details about the compression testing, what kind of compression was carried out and the direction of compression. Was there any ASTM standard followed? What is the size of the sample? 

Changes have been made to the manuscript to clarify research methods:

"The diameter of the titanium disk on which the sample was deposited was 54 mm. The overall dimensions of the obtained samples did not exceed 50x25x40 mm."

"The hardness was determined using the Vickers scale on a hardness tester HV-1000 according to GOST 2999-75. Determination of hardness and the study of the microstructure was carried out along three planes, according to the directions indicated by the arrows in the Figure 5b. Hardness tests were carried out on the same specimens on which the microstructure was studied. The hardness was measured at 5 points (Figure 5b) for each sample and the average was determined.

The compression test pieces were 10 mm in diameter and 20 mm in height. Compression tests were carried out on a universal testing machine INSTRON 5882, compression speed was 2 mm / min according to GOST 25.503-97. Compression tests were repeated three times for each type of samples".

Figure 8, which explains where 4 samples for compression testing come from when investigating 2 methods of alloying samples, has been moved to the "Methods" section and in the new version of the article is designated Figure 6.

3) In the result section alpha and beta phases should be marked in the picture. In figure 8a, There is a pore in parallel techniques and in 8b there are no pores. Was these pores were made for any purpose? or it is a defect?  If this is a defect it should also be marked in the SEM graph (Fig 6 and Fig 7). 

In the new article, in Figures 7 and 8, the phases are indicated and the pores are indicated.

Added to the new version of the article:

"Pores are undesirable in this study. The reason for their formation is the shielding gas argon, which did not have time to escape when the powder was melted."

4) In the result section table 3 shows that the compressive strength of vertical tracks is higher and followed by perpendicular. The author must explain the reason behind this is discussion section along with the direction of the applied load. How the hardness value was checked must be explained using the picture. why should hardness value change due to different direction ? 

The following information has been inserted into the "Discussion": 

"The reason for the pronounced anisotropy of properties is directional crystallization and subsequent recrystallization when the next layer is applied. Subsequent recrystallization is confirmed by the formation of grains larger than the track height. However, a more detailed explanation of the recrystallization mechanism and the dependence of properties on the direction of crystal growth requires targeted research that is planned for the future."

Changes have been made to the manuscript to clarify research methods:

"The hardness was determined using the Vickers scale on a hardness tester HV-1000 according to GOST 2999-75. Determination of hardness and the study of the microstructure was carried out along three planes, according to the directions indicated by the arrows in the Figure 5b. Hardness tests were carried out on the same specimens on which the microstructure was studied. The hardness was measured at 5 points (Figure 5b) for each sample and the average was determined."

5) Figure 9 graph should be more clear. 

In the new version of the article, the drawing is completely redesigned.

In the new manuscript, see Figure 10.

6) Overall the study must be replanned and presented in a better way. 

The manuscript was revised and a new version uploaded.

Reviewer 3 Report

This submission is in principle interesting, but it cannot be accepted for publication for several reasons:

  1. The authors write a lot about casting complex shapes, but they show only cuboid samples. An example for a complex form would be nice.
  2. What about tensile tests?
  3. The authors write that the samples contain two phases, but they do not inform how the phases were identified.
  4. The writings in Fig. 8 are too small.
  5. The colors in Fig. 9 are so similar that one cannot swee which diagram belongs to which sample.
  6. The styling of the references is not uniform.
  7. The English contains many mistakes; some sentences cannot be understood.
  8. There are undefined acronyms; units should not be in []; there are typos.

Author Response

Dear reviewer, thank you for appreciating our work.

We have done some work to improve the manuscript taking into account your comments.

  1. The authors write a lot about casting complex shapes, but they show only cuboid samples. An example for a complex form would be nice.

The following information has been added to the introduction:

"Earlier, we carried out work [26] on growing a special item of parabolic shape by combining the Selective Laser Melting and Direct Metal. Deposition." 

2. What about tensile tests?

Agree, tensile tests are just as important as compression tests. We are currently doing this work, but its results will be published in the next article. This was written in the conclusions:

"It is expedient to investigate the effect of the Ti6Al4V alloy powder material deposition track when testing samples for static tension in the future."

3. The authors write that the samples contain two phases, but they do not inform how the phases were identified.

The new version of the manuscript contains information on the method of phase identification:

"The study of the microstructure of the obtained samples showed that a mechanical mixture of α and β-phases is formed as a result of melting (Figure 7, Figure 8). Earlier we carried out work [26], in which we studied in more detail the phase composition of the samples obtained from the powder Ti-6Al-4V using similar fusion modes. The presence of two phases (α and β) in the structure is confirmed by XRD results."

4. The writings in Fig. 8 are too small.

The inscriptions in the figure are enlarged. In the new version of the article, this is Figure 9.

5.The colors in Fig. 9 are so similar that one cannot swee which diagram belongs to which sample.

The figure has been completely redone and in the new version of the article it is Figure 10. Graphic dependencies are presented for each type of sample separately.

6. The styling of the references is not uniform.

Links verified. They correspond to the sources and logic of information presentation.

7.The English contains many mistakes; some sentences cannot be understood.

English is additionally proofread. Changes have been made.

8. There are undefined acronyms; units should not be in []; there are typos

Corrections in the text were made:

"The melting of the powder was conducted according to the previously obtained mode [24]: power P = 1800 W; powder feed K = 27 g / min; laser movement speed U = 25 mm / s; displacement of the track in the layer plane Δl = 2.5 mm; track displacement in the vertical plane Δh = 0.3 mm; number of layers n = 150." 

Round 2

Reviewer 2 Report

The paper can be accepted in the present form. 

Reviewer 3 Report

The submission is now much better than the first time, however the English still needs improvement. Research is not studied in an article, but in a lab.